# Pyelonephritis in Pediatric Uropathic Patients: Differences from Community-Acquired Ones and Therapeutic Protocol Considerations. A 10-Year Single-Center Retrospective Study

**DOI:** 10.3390/children8060436

**Published:** 2021-05-23

**Authors:** Giovanni Parente, Tommaso Gargano, Stefania Pavia, Chiara Cordola, Marzia Vastano, Francesco Baccelli, Giulia Gallotta, Laura Bruni, Adelaide Corvaglia, Mario Lima

**Affiliations:** 1Pediatric Surgery Department, IRCCS Sant’Orsola-Malpighi University Hospital, via Massarenti 11, 40138 Bologna, Italy; tommaso.gargano2@unibo.it (T.G.); stefaniapavia@hotmail.com (S.P.); chiaramberle@gmail.com (C.C.); marzia.vastano@icloud.com (M.V.); mario.lima@unibo.it (M.L.); 2Specialty School of Paediatrics, Alma Mater Studiorum—University of Bologna, 40138 Bologna, Italy; francesco.baccelli2@studio.unibo.it (F.B.); giulia.gallotta2@studio.unibo.it (G.G.); laura.bruni4@studio.unibo.it (L.B.); 3School of Medicine, Alma Mater Studiorum—University of Bologna, 40126 Bologna, Italy; adelaide.corvaglia@studio.unibo.it

**Keywords:** Pediatric Urology, pyelonephritis, antibiotic therapy, antibiotic resistant, uropathogens, vesicoureteral reflux

## Abstract

Pyelonephritis (PN) represents an important cause of morbidity in the pediatric population, especially in uropathic patients. The aim of the study is to demonstrate differences between PNs of uropathic patients and PNs acquired in community in terms of uropathogens involved and antibiotic sensitivity; moreover, to identify a proper empiric therapeutic strategy. A retrospective study was conducted on antibiograms on urine cultures from PNs in vesicoureteral reflux (VUR) patients admitted to pediatric surgery department and from PNs in not VUR patients admitted to Pediatric Emergency Unit between 2010 and 2020. We recorded 58 PNs in 33 patients affected by VUR and 112 PNs in the not VUR group. The mean age of not VUR patients at the PN episode was 1.3 ± 2.6 years (range: 20 days of life–3 years), and almost all the urine cultures, 111 (99.1%), isolated Gram-negative bacteria and rarely, 1 (0.9%), Gram-positive bacteria. The Gram-negative uropathogens isolated were *Escherichia coli* (97%), *Proteus mirabilis* (2%), and *Klebsiella* spp. (1%). The only Gram-positive bacteria isolated was an *Enterococcus faecalis*. As regards the antibiograms, 96% of not VUR PNs responded to beta-lactams, 99% to aminoglycosides, and 80% to sulfonamides. For the VUR group, mean age was 3.0 years ± 3.0 years (range: 9 days of life–11 years) and mean number of episodes per patient was 2.0 ± 1.0 (range: 1–5); 83% of PNs were by Gram-negatives bacteria vs. 17% by Gram-positive: the most important Gram-negative bacteria were *Pseudomonas aeruginosa* (44%), *Escherichia coli* (27%), and *Klebsiella* spp. (12%), while *Enterococcus* spp. determined 90% of Gram-positive UTIs. Regimen ampicillin/ceftazidime (success rate: 72.0%) was compared to ampicillin/amikacin (success rate of 83.0%): no statistically significant difference was found (*p* = 0.09). The pathogens of PNs in uropathic patients are different from those of community-acquired PNs, and clinicians should be aware of their peculiar antibiotic susceptibility. An empiric therapy based on the association ampicillin + ceftazidime is therefore suggested.

## 1. Introduction

Pyelonephritis (PN) represent an important cause of morbidity in the pediatric population and accounts for 0.7% of physician office visits and 5–14% of emergency department visits by children annually [1,2,3,4].

It is important to detect PNs in order to avoid subsequent renal scarring, arterial hypertension, or end-stage kidney disease [5,6,7].

Moreover, PN needs an empiric antibiotic therapy until the result of urine cultures, which are rarely available in less than 48–72 h.

An appropriate empiric therapy requires a knowledge of the principal uropathogens and their local incidence as well as their antimicrobial susceptibility [8,9,10,11,12,13].

PNs faced by pediatric surgeons are even more insidious since they affect patients that, due to urologic disease, frequently attend hospitals and undergo invasive exams, bringing them in contact with nosocomial bacteria known to have a completely different antimicrobial susceptibility if compared with those of febrile community-acquired UTI’s (CAUTI).

The aim of the study is to demonstrate differences in etiology between CAUTIs and PNs in uropathic patients, to evaluate the antimicrobial susceptibility of pyelonephritis of our cohort of patients affected by vesicoureteral reflux (VUR), and to infer a proper antimicrobial empiric therapy able to cover the majority of uropathogens.

## 2. Materials and Methods

After ethical committee approval (CHPED-05-20-URO), a retrospective study was conducted.

We analyzed clinical records of PNs in uropathic patients (VURs) from 2010 to 2020 admitted to our Pediatric Surgery Department and PNs in patients not affected by VUR or other uropathies admitted in the Pediatric Emergency Unit, taking into account age of the patients, symptoms, urine cultures in terms of modality of sampling, and antibiogram’s results.

A PN was defined by evidence of urinary tract infection from urinalysis and culture along with signs and symptoms suggesting upper urinary tract infection (fever > 38 °C, chills, flank pain, nausea, and vomiting); all patients underwent abdomen ultrasound in order to detect signs, such as abnormal echogenicity of the renal parenchyma, that could confirm the hypothesis of PN.

We considered positive urine culture suggestive of pyelonephritis as the ones that showed an antibiogram with the subsequent characteristics:
Growth of ≥100,000 colony forming units (CFU)/mL of one uropathogen or growth of ≥100,000 CFU/mL of a uropathogen and ≤50,000 CFU/mL of a second one.Positivity for clinically relevant uropathogens.We excluded antibiograms suggestive for contamination of urine samples; the exclusion was made on the basis of the subsequent parameters:Growth of ≥50,000 CFU/mL of a second uropathogen or >2 organisms isolated.Positivity for irrelevant uropathogens.


We considered the following as relevant uropathogens [14]: Escherichia coli, Klebsiella spp., Proteus spp., Enterobacter spp., Citrobacter spp., Serratia marcescens, Staphylococcus saprophyticus, Enterococcus spp., Streptococcus agalactiae, Pseudomonas aeruginosa, and Staphylococcus aureus.

The subsequent were considered irrelevant uropathogens [2]: *Lactobacillus* spp., *Corynebacterium* spp., and coagulase-negative Staphylococci.

In the Pediatric Surgery Department, urine samples of the patients were obtained with a clean-voided specimen in children who were toilet trained and with a sterile collection bag in those not toilet-trained, while in the Pediatric Emergency Department, the collection of urine sample was performed either in the same way or via bladder catheterization depending on the pediatrician’s preference.

We analyzed the efficacy of the principal antibiotic molecules of pediatric interest and then, guided by the results of the study, we formulated an empirical therapeutic protocol that can be as effective as possible in the majority of PNs in this particular population of pediatric patients.

Data on protocols efficacy were analyzed via inferential statistics for proportions, and a *p*-value <0.05 was considered statistically significant.

All descriptive data are expressed as mean ± standard deviation.

## 3. Results

### 3.1. Not Uropathic Population (Not VUR)

From 2010 to 2020, we recorded 112 PNs, 61 (54.5%) in males and 51 (45.5%) in females.

The mean age of not VUR patients at the time of PN episode was 1.3 ± 2.6 years (range: 20 days of life–13 years). Twenty-two (19.6%) of the urine samples collected were obtained via urethral catheterization vs. 90 clean-voided specimens or those collected via sterile bag.

Data collected showed that almost all the urine cultures, 111 (99.1%), isolated Gram-negative bacteria and rarely, 1 (0.9%), Gram-positive ones.

The Gram-negative uropathogens isolated were *Escherichia coli* (97%), *Proteus mirabilis* (2%), and *Klebsiella* spp. (1%) (Figure 1).

The only Gram-positive bacteria isolated was an *Enterococcus faecalis*.

As regards the antibiograms, 96% of not VUR PNs responded to beta-lactams, 99% to aminoglycosides, and 80% to sulfonamides (Figure 2).

### 3.2. Uropathic Population (VUR)

From 2010 to 2020, we recorded 58 PNs in 33 patients affected by VUR, 19 (57.6%) males and 14 (42.4%) females.

The mean age of VUR patients at the PN episode was 3.0 years ± 3.0 years (range: 9 days of life–11 years) and the mean number of episodes per patients was 2.0 ± 1.0 (range: 1–5).

A total of 16 (48.5%) patients were affected by bilateral VUR vs. 17 unilateral VUR.

Five (15.2%) patients were affected by third-grade VUR and treated with a Deflux implant procedure, while the other 28 (84.8%) were affected by fourth- or fifth-grade VUR that was treated with Cohen ureteral reimplantation; at the same time of both the procedures, a circumcision was performed in male patients.

All patients, when admitted with our department for PN, were already in antibiotic prophylaxis with amoxicillin at a dosage of 25 mg/Kg once a day.

Nine (27%) PNs were recorded after the surgical treatment vs. 24 (73.0%) before surgery. 

Data collected confirmed the well-known prevalence of Gram-negative bacteria even in our PNs: 83% Gram-negatives vs. 17% Gram-positives.

The most frequent Gram-negative uropathogens were *Pseudomonas aeruginosa* (44%), *Escherichia coli* (27%), and *Klebsiella* spp. (12%), which together count for the 83% of all Gram-negatives PNs, while 90% of Gram-positive PNs were caused by *Enterococcus* spp. (Figure 1).

As regards the previously mentioned pathogens, Figure 3 shows the antimicrobial susceptibility recorded during the 10 years of the study.

*P. aeruginosa* showed a complete response to cephems (100%), carbapenems (100%), and aminoglycosides (100%). We remind that *P. aeruginosa* is constitutive resistant to sulfonamides. No carbapenem-resistant *Pseudomonas aeruginosa* (CRPA) was isolated.

*E. coli* showed a great response to aminoglycosides (92%), but less to cephems (69%) and poor to penams and sulfonamides (54%). It is important to mention the fact that the 92% of response to carbapenems was due to a carbapenemase-producing *Escherichia coli* (CP-Ec) isolated.

*Klebsiella* spp. showed a complete response to carbapenems and aminoglycosides (100%), a good response to penams and cephems (83%), but poor response to sulfonamides (50%). No *Klebsiella pneumoniae* carbapanemase-producing (KPC) was isolated.

*Enterococcus* spp. showed a 33% response to penams, being entirely susceptible to vancomycin (100%).

Considering all the PNs analyzed, the antimicrobial susceptibility distribution is shown in Figure 2 and Figure 4A: a poor response was recorded as regards sulfonamides, but a good response considering beta-lactams and aminoglycosides.

As regards the single molecules of beta-lactams family, we analyzed the antibiograms of our PNs, and the results are synthetized in Figure 4B,C.

Ampicillin (AMP) showed a poor efficacy, especially if compared with the highly performant piperacillin/tazobactam (TZP) (62% of efficacy) and ceftazidime (CAZ) (67%). Meropenem (MEM) showed a good efficacy that decreases only when faced with Gram-positive (not responsive to carbapenems) PNs.

Taking in account the cephems family, the pathogens isolated were responsive only to third- and fourth-generation ones; therefore, considering the high incidence of *P. aeruginosa*, we did not take into consideration ceftriaxone (CRO, to which *P. aeruginosa* is constitutively not responding) as representative of this class, opting for CAZ instead.

On the basis of the data described so far, we conceived two empiric therapeutic protocols that could be administered to urologic patients affected by PN (Figure 4).

The first protocol analyzed consisted in the association of a Gram-positive active penam (AMP) with a third-generation cepham (CAZ): this had a success rate of 72.0% (Figure 5).

The second protocol studied consisted in the association of a Gram-positive active penam (AMP) with an aminoglycoside (amikacin, AMK): this had a success rate of 83.0% (Figure 5).

There is no statistically significant difference between the two protocols (*p* = 0.09).

## 4. Discussion

PN is a serious urinary tract infection (UTI) due to the potential risk of subsequent renal scarring, arterial hypertension, or end-stage kidney disease.

In the case of the pediatric surgeons’ patients, PN affects a kidney already potentially compromised by urological diseases, conditions that also limit the choice of antibiotics.

We also have to be aware of the fact that uropathic patients frequently attend hospitals and undergo invasive exams, both conditions that expose these children to nosocomial bacteria well known to be acquiring broad resistance to the most frequently used antimicrobial agents [15].

Furthermore, hospitals should be aware of the bacteria circulating in their geographical area as well as their antibiogram; this because the first therapeutic approach to PN is necessary empiric [16,17].

Our study highlights some fundamental concepts of UTI antibiotic therapy in uropathic patients and adds some etiological elements that show how a PN in a non-uropathic patient can be a nosologically completely different entity from that of uropathic patients.

However, to better understand what is shown by our data, it is first necessary to briefly address a concept recently introduced in medicine: contrary to the knowledge handed down from the past, the urinary tract is not sterile.

Emerging evidence shows that the bacteria inhabit many sites of the body, including the urinary tract, and may have some role in maintaining urinary health: in fact, the term “urinary microbiota” is now increasingly widespread in literature, referring precisely to the concept just exposed [18,19,20].

The precise role of the urinary microbiota in human urinary health is still the subject of extensive studies and requires further investigation, but its usefulness has been demonstrated, for example, in the prevention or diagnostics of bladder cancer, as well as being the cause of various diseases and symptoms of the urinary tract other than simply infectious disease [21,22,23,24].

At the moment, urologists rarely need to consider bacteria beyond their role in infectious disease; however, in the future, urinary microbiota will enter in the daily medical practice, providing an opportunity to predict the risk of developing certain urological diseases and enabling the development of innovative therapeutic strategies.

Considering what is mentioned above, it appears plausible that medical interventions on the urinary tract can modify the resident bacterial flora.

Pediatric uropathic patients undergo invasive exams that requires catheterization as well as, in case of VUR, require long-term antibiotic prophylaxis: the first determines colonization by nosocomial bacteria and the second gives rise to selective pressure producing antibiotic-resistant strains.

This is why we decided to investigate the etiology of PNs in uropathic patients and compare them with the ones acquired in the community; to do that, we considered VUR-affected patients as an example of a uropathic patient.

Figure 1A shows how, in terms of Gram stain, the uropathogens between CAUTIs and VUR PNs are absolutely similar, even if the second has a slightly higher incidence of Gram-positive PNs, which we cannot exclude being a result of a contamination of the urine samples.

Figure 1B–E is a graphical representation of the concept nosocomial microorganism colonization mentioned above.

In fact, while CAUTIs are almost complete prerogative of *Escherichia coli* (99.1%), the VUR group PNs show a wider range of responsible uropathogens: the most frequent one is *Pseudomonas aeruginosa* (44%), followed by *Escherichia coli* (27%) and *Klebsiella* spp. (13%).

The aforementioned bacterial flora reflect the main hospital-resident pathogens, and this is particularly worrying in a historical era in which antibiotic resistance is increasingly reported [25,26,27,28,29,30,31].

Figure 2 shows the effect of selective pressure of antibiotics. In fact, CAUTIs show an overall good response profile to antibiotics, information derived from antibiograms: beta-lactams with efficacy in 96% of PNs, aminoglycosides in 99%, and sulfonamides in 80%, the latter is not a novelty since sulfonamides-resistant *E. coli* are known to be increasing [32,33,34,35,36]. Paying attention to VUR PNs, we can notice a dramatic decrease in antimicrobial sensitivity: only 78% of PNs were responsive to beta-lactams, 71% to aminoglycosides, and just a worrying 29% to sulfonamides.

The reasons of this important and alarming difference between the two populations are to be found, as previously mentioned, in the effect that antibiotic prophylaxis may have determined in selecting resistant strains [15,37] but also in the colonization in these particular children by nosocomial multidrug-resistant strains with which they came into contact during invasive investigations in the course of their diagnostic and therapeutic process.

We can therefore conclude that CAUTIs and PNs in uropathic patients are completely different in terms of uropathogens and antibiotic sensitivity, and pediatricians as well as urologists should be aware of this important concept.

At this point, we focused on VUR group PNs and wondered what the best empirical therapeutic approach is to be as confident as possible of having covered the main responsible uropathogens.

As mentioned above, our study shows that the main pathogens involved are Gram-negatives and, of these, the most represented is *Pseudomonas aeruginosa*.

This consideration allows us to exclude CRO, a third-generation cephalosporin frequently used also in our center, from clinical practice because it is not active against *Pseudomonas* spp.

Figure 4B,C shows the beta-lactams spectrum of efficacy in our cohort of patients: considering the most frequently used and available antibiotic molecule in pediatrics, TZP and CAZ appeared to be the best performing with a slightly higher efficacy recorded for CAZ.

AMP showed a poor efficacy in our PNs, but we have to keep in mind that it can be precious to extend the spectrum of other molecules to Gram-positive strains.

Our data also showed that, contrary to the past, sulfonamides no longer represent a valid therapeutic choice in uropathic children with PN.

A delicate issue concerns aminoglycoside: this class of antibiotics showed excellent efficacy against the isolated pathogens in our PNs; however, uropathic patients can be affected by chronic kidney damage of various grade related to the uropathy itself. Considering the nephrotoxicity of aminoglycosides, it is clear that the indication for their use is extremely limited; we therefore tried to avoid their involvement in our therapeutic protocol proposal.

On the basis of the previous analysis, we therefore identified two empiric therapeutic protocols, which consist in the association of two antibiotics each due to the necessity to cover both Gram-negative and Gram-positive microorganisms: AMP + CAZ and AMP + AMK.

AMP was chosen, as previously stated, as a penicillin active against Gram-positives such as *Enterococcus* spp.

AMK was chosen because it is frequently used in our center, but it can be replaced by gentamicin as well.

Revising the antibiograms of our PNs, the two protocols respectively showed a success rate of 72.0% and 83.0%, but statistical analysis did not show a significant difference between the two approaches’ efficacy (*p* = 0.09).

These data are extremely important because they allow us to freely decide which of the two protocols to adopt, and, by opting for AMP/CAZ, we can eliminate the aminoglycosides from the antibiotic therapy of PN, avoiding the risk of worsening renal function due to the well-known adverse effects of these drugs.

As regards the posology regimen, the EAU/ESPU Guidelines [38] present dose and duration of various antibiotic therapies; our protocol is suggested to be administered parenterally for at least 7 days and then shifted to oral therapy (if decreasing of inflammatory laboratory values and clinical improvement) to complete a 15-day therapy with antibiotic agents chosen on the bases of the antibiogram. The posology of our empiric approach is the following:AMP: 100–200 mg/Kg divided in 3–4 doses daily.CAZ: 100–150 mg/Kg divided in 2–3 doses daily.

In agreement with our pediatric infectious disease specialists, in case of Gram-negative uropathogens nonresponding to the AMP/CAZ protocol, it is preferable to switch directly to carbapenems rather than adding an aminoglycoside.

If therapy failure is due to Gram-positive bacteria, the AMP/CAZ protocol will be suspended, and vancomycin (VAN) administered. About this issue, we are aware that AMP covered only 33% of the Enterococci isolated, the principal Gram-positives involved, but it is not sufficient to justify an introduction of VAN as first-attempt therapy considering the higher prevalence of Gram-negatives; however, at the same time, a 17% incidence of Gram-positive PNs requires an empiric coverage and AMP seems to be a good compromise.

Although reports of carbapenemase-producing organisms are growing in literature, in our experience we have isolated only one CR-Ec in 10 years.

We are aware of the limitations that this study brings with it. First of all, the risk of drawing erroneous conclusions from contaminated urine cultures due to the method of collecting urine samples. However, this cannot justify the extremely significant incidence of nosocomial bacteria in this particular population. Moreover, for the reasons discussed above, the colonization of these children by the isolated bacteria appears absolutely probable. Finally, most of the urine cultures of the not VUR group were obtained with the same collecting method of the VUR group, nevertheless, the proportions of bacteria like *P. aeruginosa* and *Klebsiella* spp. identified in this population are absolutely not comparable to those of the VUR group; this, from our point of view, strengthens the idea of the reliability of our data.

The other important limitation of this study is the low numerosity on which it is based. This prevents us to extend our conclusions to other centers, but, confirming the validity of our conclusions, it is encouraging that the guidelines of the European Society of Pediatric Urology confirm our suggestions [38].

However, we believe that the present paper is sufficiently valid to stress the concept that pyelonephritis in uropathic patients should be considered as an entity apart from those acquired in the community and requiring a more aggressive antibiotic therapy.

Certainly, multicenter studies are needed to confirm our hypotheses, but we are confident that our data will motivate the reproduction of this study in other centers.

## 5. Conclusions

PN in uropathic patients should be considered apart from CAUTIs in terms of both uropathogens involved and antibiotic sensitivity.

Considering the potential pre-existing kidney damage due to uropathy itself, aminoglycosides should be avoided.

The experience reported in this study led us to adopt an empirical antibiotic protocol based on the association of AMP and CAZ; further studies are needed to confirm our data.

## Figures and Tables

**Figure 1 children-08-00436-f001:**
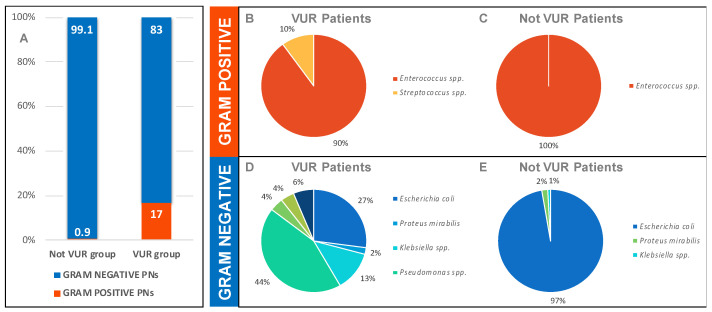
(**A**) Gram prevalence on PN between 2010 and 2020 in VUR and not VUR groups (%), (**B**) and (**C**) Gram-positive uropathogens isolated in VUR and not VUR PNs between 2010 and 2020; (**D**,**E**) Gram-negative uropathogens isolated in VUR and not VUR PNs between 2010 and 2020.

**Figure 2 children-08-00436-f002:**
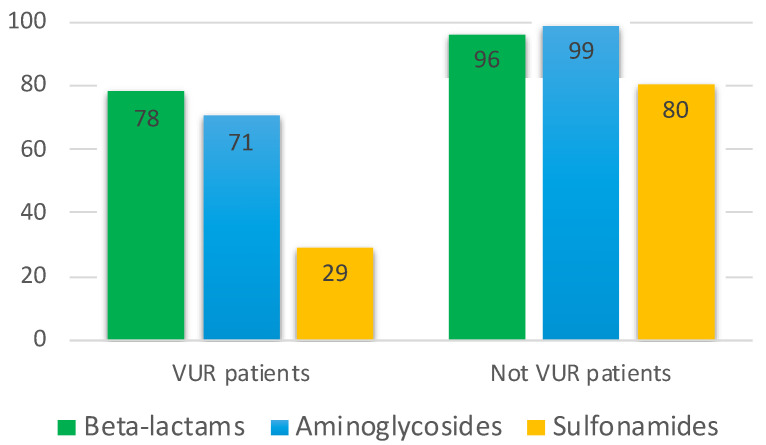
Efficacy of classes of antibiotics of principal interest in pediatrics showed by antibiograms in patients affected by PN in the VUR and not VUR groups (%).

**Figure 3 children-08-00436-f003:**
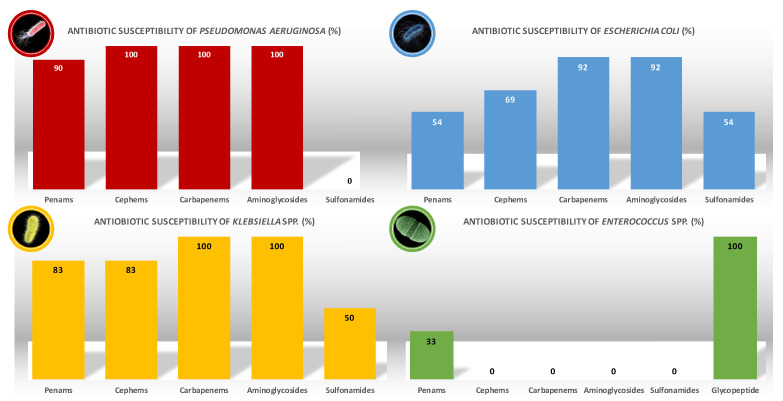
Antibiotic susceptibility of principal uropathogens isolated in uropathic patients.

**Figure 4 children-08-00436-f004:**
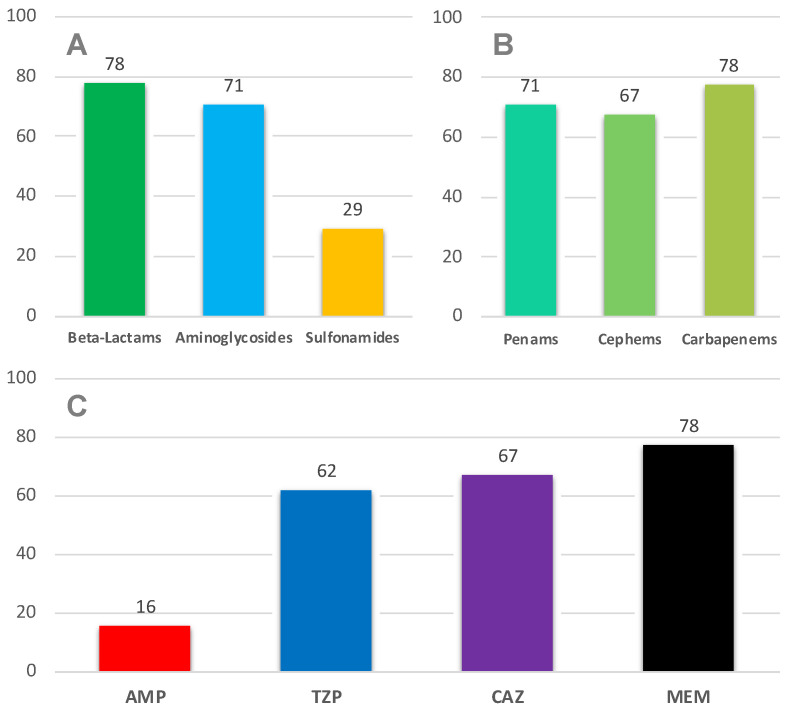
Susceptibility to principal antibiotics of pediatric interest (**A**) in VUR PNs (%); beta-lactams susceptibility (**B**) in VUR PNs (%); beta-lactam molecules susceptibility (**C**) in VUR PNs (%). AMP: ampicillin, TZP: piperacillin/tazobactam, CAZ: ceftazidime, MEM: meropenem.

**Figure 5 children-08-00436-f005:**
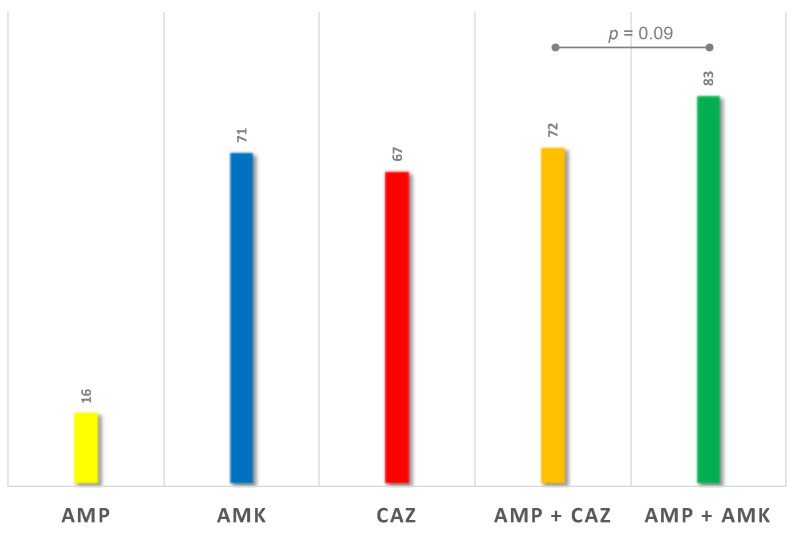
Efficacy of protocol proposals (%). AMP: ampicillin, AMK: amikacin, CAZ: ceftazidime.

## Data Availability

The data presented in this study are available on request from the corresponding author. The data are not publicly available due to privacy restrictions.

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
