# Peer review of "Pyelonephritis in Pediatric Uropathic Patients: Differences from Community-Acquired Ones and Therapeutic Protocol Considerations. A 10-Year Single-Center Retrospective Study"

_children, 2021, doi:10.3390/children8060436_

Round 1

Reviewer 1 Report

Dear Authors!

The topic of your study is relevant - congenital malformations do not only predispose children to febrile UTIs, this group of patients is also more endangered in regards to recurrence or subsequent renal damage. An antibiotic prophylaxis might be given to many, further predisposing them for breakthrough infections with aberrant bacteria. The manuscript is clearly structured, the data is presented with attention to detail. 

The most relevant concern about this paper is the - discussed - fact, that these microbial patterns are only relevant for the authors institution. A general recommendation is not possible. Furthermore, there is no direct comparison to the bacterial spectrum in fUTIs in children without congenital malformations of the urogenital tract. Methodologically, bag urine grants for contamination, partly the astonishingly high proportion of Pseudomonas Aeruginosa might be also partly due to the sampling methodology. There are no clinical criteria for pyelonephritis mentioned, potentially based on the retrospective design based on culture results.

Also, there is only little of the vast body of available literature discussed or cited. Not only in regards to bacterial causes of fUTIs but also regarding the application of antibiotics, their duration etc. 

The only general conclusion to be drawn is that every hospital should review their urine culture results to adapt the empiric therapy - which is nothing new.

Author Response

Dear Reviewer,

Thank you for your precious comments that enhanced the scientific quality of our paper.

In the following lines I will answer to your kind requests.

  1. […] The most relevant concern about this paper is the - discussed - fact, that these microbial patterns are only relevant for the authors institution. A general recommendation is not possible.

We are aware of it, therefore rather than propose recommendations, in the revised paper is now stated why we think this could be a common pattern to other urological units (colonization by nosocomial bacteria due to medical interventions and effects of antibiotic prophylaxis) and we encourage multicenter studies. Lines: 299-302, 306-307.

  1. Furthermore, there is no direct comparison to the bacterial spectrum in fUTIs in children without congenital malformations of the urogenital tract.

We now added the comparison, thank you for the suggestion.

  1. Methodologically, bag urine grants for contamination, partly the astonishingly high proportion of Pseudomonas Aeruginosa might be also partly due to the sampling methodology.

We are aware of the limits of the collecting method. By the way, adding a not-VUR group, where most of the urine sample were obtained with the same method, helped us confirming the validy of our data. In fact, if the proportion of Pseudomonas is to be attributed solely to the collection method, then it should be similar to the not-VUR group, which is not the case.

  1. There are no clinical criteria for pyelonephritis mentioned, potentially based on the retrospective design based on culture results.

They are now reported in Material and Methods section. Lines: 62-66.

  1. Also, there is only little of the vast body of available literature discussed or cited. Not only in regards to bacterial causes of fUTIs but also regarding the application of antibiotics, their duration etc.

We enhanced the quantity of literature citated in the text.

  1. The only general conclusion to be drawn is that every hospital should review their urine culture results to adapt the empiric therapy - which is nothing new.

Adding the not-VUR group, the discussion enriched showing the etiologically difference between PNs in uropathic patients and the ones in patients without congenital malformation of the urinary tract.

We hope we answered to all your kind comments, and we thank you again for your suggestions.

Reviewer 2 Report

  1. This is an important and novel idea to analyze the differences in uropathogens presenting in children with uropathy. On my review, there are relatively few studies discussing the difference in pathogens between patients with reflux and those without. It stands to reason that the increased prevalence of Pseudomonas (shown in other studies - Bitsori, et. al, 2012 J Urol) is a produce of recurrent infections with increase in antibiotic resistance and use of antibiotic prophylaxis.
  2. It would be interesting, and in my mind, beneficial to compare the uropathogens isolated at your institution over this time period in those without reflux compared to those with reflux to illustrate the differences.
  3. Is antibiogram being used synonymously with urine culture? I found this confusing.
  4. Although it is fair to present your antibiotic algorithm for these patients, it is an overextension of your results to recommend this therapy to other populations, especially with the low numbers in your series.
  5. Figure 4 was confusing and should be explained in text.
  6. There is no mention of patient characteristics - we must know the number of episodes of pyelonephritis over this time, presence or absence of antibiotic prophylaxis, age at infection, gender, circumcision status.
  7. Limitations must be addressed - lack of generalizability, small numbers

Author Response

Dear Reviewer,

Thank you for your precious comments that enhanced the scientific quality of our paper.

In the following lines I will answer to your kind requests.

  1. It would be interesting, and in my mind, beneficial to compare the uropathogens isolated at your institution over this time period in those without reflux compared to those with reflux to illustrate the differences.

Thank you for the suggestions, we added the comparison.

  1. Is antibiogram being used synonymously with urine culture? I found this confusing.

Sometimes it erroneously was, but we now corrected.

  1. Although it is fair to present your antibiotic algorithm for these patients, it is an overextension of your results to recommend this therapy to other populations, especially with the low numbers in your series.

We agree and it is now explained in the limitation section of the discussion. Even the title was changed highlighting that these are more considerations than recommendations.

  1. Figure 4 was confusing and should be explained in text.

We realized some figures where confusing, therefore we simplified the figures and better explained them in the text.

  1. There is no mention of patient characteristics - we must know the number of episodes of pyelonephritis over this time, presence or absence of antibiotic prophylaxis, age at infection, gender, circumcision status.

The patient characteristics are now reported in the results section.

  1. Limitations must be addressed - lack of generalizability, small numbers

A section dedicated to Limitations was added at the end of the Discussion paragraph. Lines: 302-315.

We hope we answered to all your kind comments, and we thank you again for your suggestions.

Round 2

Reviewer 1 Report

While the main issue remains - that this is relevant mostly to one center as far as the exact germ spectra and therapeutic recommendation are concerned - the authors incorporated more data on the control group which makes this study much more interesting to read. 

Also, limitations and methodological issues (bag urine) are discussed in more detail, reflecting also more of the relevant literature in this field. 

Although I am convinced that the use of bag urine influences the incidence of false positive cultures (in both groups!) the point made by the authors (that there is still a difference between the groups) is well taken and the limitations of this conclusion are displayed now in a cautious way. 

Therefore I congratulate you to the effort made and will recommend this paper for publication.